# Datasets of Groundwater Level and Surface Water Budget in a Central Mediterranean Site (21 June 2017–1 October 2022)

Marco Delle Rose * and Paolo Martano

Institute of Atmospheric Sciences and Climate, National Research Council of Italy, 73100 Lecce, Italy
* Correspondence: m.dellerose@le.isac.cnr.it

**Abstract:** This note makes available five years of data gathered in a measurement site equipped with a micrometeorological station and two monitoring wells. Series of data of hydrological and atmospheric variables make it possible to estimate the flux of water across the atmosphere-land interface and to calculate the water budget, which are crucial topics in climate and environmental sciences. The water-table measures began during 2017, one of the driest years of the whole instrumental period of climate history for the Central Mediterranean. Data from the micrometeorological station have been used to construct two more datasets of daily and monthly totals of different terms of the surface water budget, from which the net infiltration has been estimated. An apparent decreasing trend characterizes both the data time series of groundwater level and estimated infiltration in the considered period.

**Keywords:** hydrology; precipitation; net infiltration; evapotranspiration flux; soil water balance; shallow aquifer; well hydrograph; Apulia; Italy

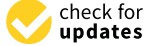



## 1. Summary

Datasets including groundwater level, soil moisture, and meteorological variables make it possible to estimate fluxes of water and energy across the atmosphere–land interface and to calculate water budgets [1–5]. Given their fundamental importance in climate and environmental sciences, series of data of hydrological and atmospheric variables taken at specific sites are becoming increasingly freely shared [6–8].

Several regions in the world, including the Euro-Mediterranean, were affected by declining groundwater resources in recent decades [9,10]. Drought events linked to increases of temperature, in combination with decreasing precipitation in the summer months, have been indicated among the main causes [11,12]. From July 2016 to June 2017 western-central Europe was affected by the most severe meteorological drought at the continental scale since at least 1979 [13]. Winter 2016/17 was drier than normal in the Central Mediterranean, with precipitation totals reaching only 60% of normal precipitation [14]. In 2017, Italy experienced the driest year of the whole instrumental period of its climate history [15].

Since June 2017, the Institute of Atmospheric Sciences and Climate of the National Research Council of Italy (ISAC-CNR) has taken measures of water-table depth of a shallow karst aquifer (the Miocene Aquifer of Central-Eastern Salento, hereinafter referred as MACES) located in the Salento Peninsula (Apulia region, Italy). The measures are taken from two wells located several hundred meters from the ISAC-CNR Micrometeorological Station (IMS), which has been in operation since June 2002 [16–18]. As a whole, wells and station constitute a measurement site that allows direct comparison between hydrological and weather data. Due to their climate sensitivity and subsurface heterogeneity, the karst systems located in dry lands require particular attention when it comes to processing data

related to the exchanges between air, soil and groundwater [19,20]. Series of data from this site have been used to interpret the local water consumption during the lockdown for COVID-19 [21]. Datasets of IMS were previously used to calculate recharge time and specific yield of the regional deep aquifer [22,23]. This note provides and makes available five hydrological years (from 1 October to 30 September in the Northern Hemisphere) of gathered data with the aim of promoting open science and fostering cooperation by making related descriptions and insights shareable.

## 2. Data Descriptor

### 2.1. Geophysical Setting

Located in the Central Mediterranean, the Salento Peninsula constitutes a lowland plateau (maximum altitude 200 m) surrounded by the Ionian Sea to the West and South and by the Adriatic Sea to the East. Annual precipitations are usually between 600 and 700 mm and less than Italian average because of the low orography and the residual blocking effect of the Apennine Chain (150 km NW of Salento), for the incoming cyclonic moisture flux coming from the Atlantic Ocean, generally accompanied by southerly winds. Moderate northerly winds typically blow during the daytime in high pressure conditions, with clear skies and strong insolation. The warm/dry season (April–September) is characterized by temperatures that can reach 40 °C and more, and reduced precipitations, while the cold/wet season (October–March) by mild autumn–winter temperatures (generally above zero even at night) and enhanced precipitation. In recent years, there has been a generalized tendency to an increase in more intense, brief precipitation events during the spring–summer months [22].

Groundwater resources of the Salento Peninsula are contained in a complex system of aquifers, consisting of a main regional reservoir, the Cretaceous Aquifer (CA), and several shallow aquifers [24–26]. Among these latter, MACES is the most important in terms of stored groundwater volume. It is supplied by meteoric water and two Plio-Quaternary aquifers, and extensively exploited for irrigation and domestic purposes through several licensed and unlicensed wells. In the warm/dry season, MACES is recharged by irrigation water withdrawn from the CA (more hydrogeological features are given in Appendix A).

### 2.2. Site and Surface Data

The measurement site is placed in a suburban area 5 km south-west from Lecce (Figure A1), within the Ecotekne Campus of the University of Salento, with mixed local vegetation including Mediterranean shrubs, pines and olive groves, scattered buildings, and quarried surfaces (Figure 1). Here, the MACES water table lies at a depth of 20–22 m below the ground, in good agreement with literature data [21,27].

The IMS [16,17] has been a data provider for the Hymex (Hydrological Mediterranean Experiment) project's long term campaign [28]. It is mainly devoted to water and energy surface–atmosphere transfer, and is composed of two complementary systems, collecting data based on half-hour averages: (1) a 16 m height mast equipped with a fast response eddy covariance system and basic meteorological data; (2) a micrometeorological station that also collects standard meteorological and soil surface data in a dedicated data logger. The mast has a typical flux footprint fetch in the order of hundreds of meters [22,29], thus taking contributions from the campus and some immediately surrounding vegetated areas. Both data files from the two systems are stored in local memory and then transferred to the web data base once per day through UMTS connection system [18]. The collected data of precipitation, real evapotranspiration, and soil moisture also allow the estimation of the net infiltration as described in Section 3, together with more details about measurements and sensors.

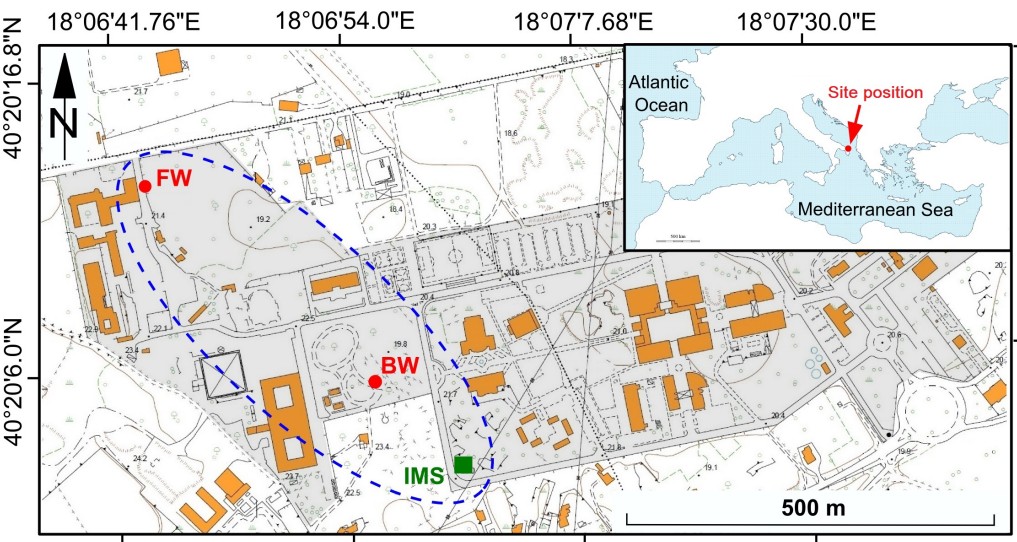

**Figure 1.** Map of the measurement site (topography from Lecce Province Technical Map at a scale of 1:5000). IMS, ISAC-CNR Micrometeorological Station; FW, Fiorini Well; BW, Benessere Well. Ecotekne Campus in gray (Apulia region, Italy).

### 2.3. Groundwater Level Data (Manual Measurements)

The two monitoring wells of the measurement site (named Fiorini, FW, and Benessere, BW, Figure 1) were hand dug about 25 m deep one century ago, to provide water for agriculture and rural settlements. The wells are easily identifiable by their stone well heads (Figures A3 and A4). Based on the provincial technical map at a scale of 1:5000, the height of the well head is 22.5 for FW and 24.7 m a.s.l. for BW. Google Earth geographic coordinates are: 40°20′11.93″ N, 18°06′43.09″ E, for FW; 40°20′03.50″ N, 18°06′55.30″ E, for BW.

Manual measurements of groundwater levels (as water-table depth below the well head, see Figure A5) began in June 2017 for FW, and in June 2018 for BW. They have been taken using a portable electronic water-table meter (Boviar GST-FR100), with an average frequency of one measure about every three days for both wells (for more details see Appendix B). The frequency of measurement was established during the first months of the monitoring aiming to obtain relatively detailed well hydrographs. However, during some rainy events, the measurements were made more frequently in order to better document the rise in the groundwater level. Despite the laboriousness of the field operations, the use of a water-table meter makes it possible to overcome problems related to the lack of research funding and the reliability of measurements in cases of equipment malfunction [30–32]. Well hydrographs contain valuable information on aquifer responses to both natural and human stresses [33].

### 2.4. Data Records

The dataset of the water-table depths taken from 21 June 2017 to 1 October 2022 is supplied as a text file in the Zenodo repository (see "S1.txt" in https://zenodo.org/record/7572140). The columns represent the following items:

- date (dd/mm/yy)
- water-table depth measured in FW (m)
- water-table depth measured in BW (m).

There are a few gaps in the dataset, because sometimes adverse weather conditions prevented measurement for one of the two wells. The curves of the variations of the groundwater head in the monitored wells are shown in Figure 2.

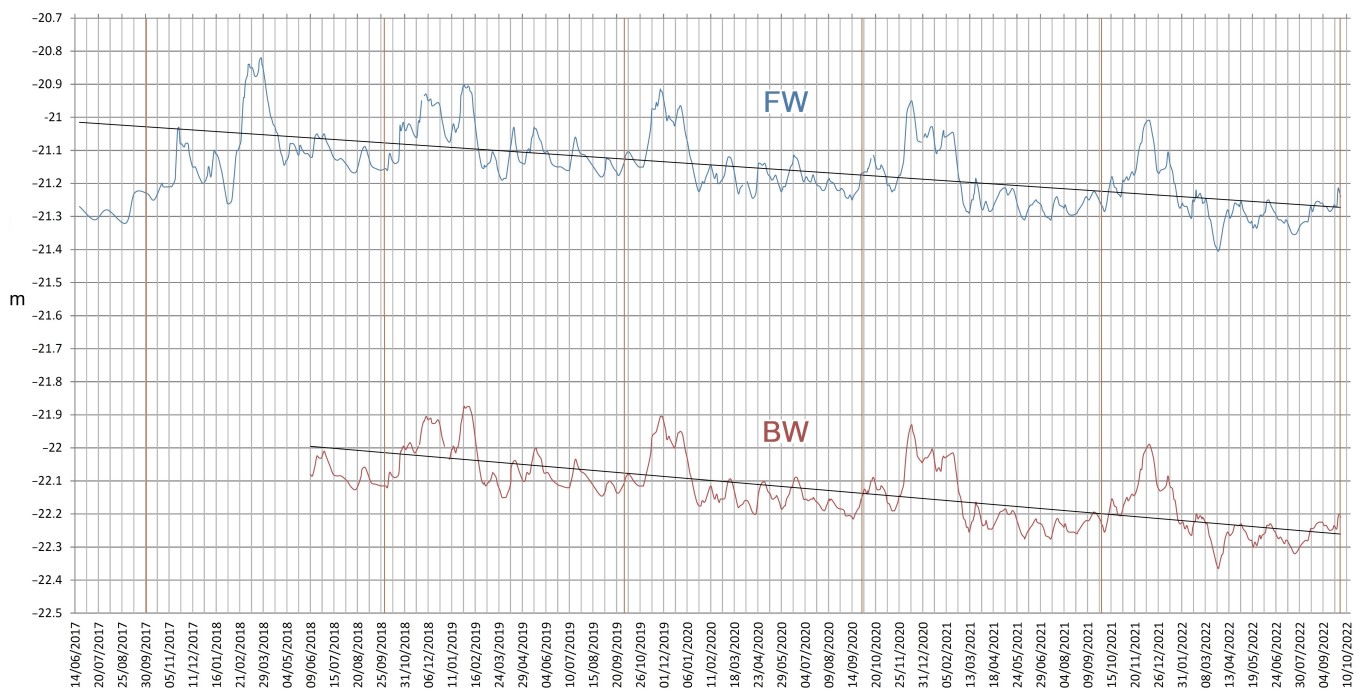

**Figure 2.** Smoothed well hydrographs with linear regression (water-table depth in meters below the well head; see Figure A5). Hydrological years are delimited; FW, Fiorini Well; BW, Benessere Well.

For FW, the measure of the water-table depth started in the middle of the warm/dry season (June 2017) of the 2016–2017 hydrological year. A year later (June 2018), monitoring of BW began. Note that the two lines fit almost perfectly. Only in very few cases were the daily variations in FW and BW of the opposite sign (see "S1.txt" in https://zenodo.org/record/7572140). A general negative trend from the hydrological year 2017–2018 to 2021–2022 is apparent. At the end of the 2021–2022 hydrological year (June-September 2022), the groundwater level in FW was nearly coincident with that of June–September 2017.

The surface water budget datasets are available in the Zenodo repository (see "S2.txt" and "S3.txt" in https://zenodo.org/record/7572140) as text files of daily/monthly data. The columns represent the following items:

- date (Julian day or fraction of year representing the end of each month)
- precipitation (daily/monthly total, m)
- real evapotranspiration (daily/monthly total, m)
- soil moisture 2 cm (beginning of the day/month, $m^3/m^3$)
- soil moisture 2 cm (end of the day/month, $m^3/m^3$)
- soil moisture 35 cm (beginning of the day/month, $m^3/m^3$)
- soil moisture 35 cm (end of the day/month, $m^3/m^3$)
- estimated net infiltration (daily/monthly total, m)
- percentage of validated data (daily/monthly total)
- corrected real evapotranspiration (daily/monthly total corrected by percentage of validated data, m)
- corrected net infiltration (daily/monthly total corrected by percentage of validated data, m).

The corrected monthly net infiltration values for the MACES are shown in Figure 3, where still possible negative values have been set to zero to properly represent the groundwater contribution (see also Section 3). Other data are plotted in Figures A6 and A7 (Appendix C).

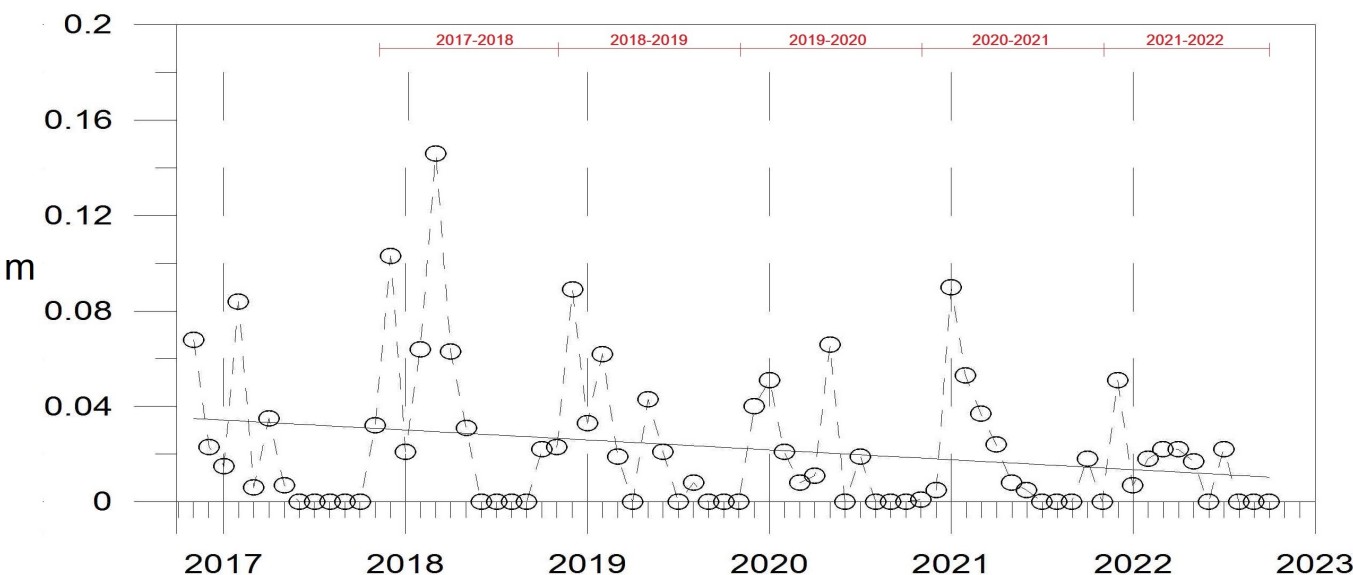

**Figure 3.** Estimated monthly net infiltration in meters with linear regression (continuous line). The dashed line is intended only to help viewing. The considered hydrological years are delimited in the upper side.

## 3. Methods

### 3.1. Groundwater Level Measurements

To collect groundwater levels data, reference points were fixed on the stone casing of each monitored well. A rod was placed on the well head to reference the depth read by the water-table meter (Appendix B, Figures A3 and A5). In this way, it was possible to measure the depth with an accuracy of 0.5 cm. Furthermore, this system allowed the use of several operators without affecting the reliability of the data.

### 3.2. Surface Measurements

#### 3.2.1. Telescopic Mast (Top Level 14 m above Surface)

The Eddy covariance system (measurements of turbulence, heat, and evapotranspiration fluxes) is composed by one Solent–Gill 20 Hz ultrasonic anemometer and one Campbell Kh20 Krypton hygrometer. One thermohygrometer allows standard air temperature and humidity measurements (Rotronic MP100). Data are firstly collected in a dedicated notebook that calculates the half hour eddy covariance statistics rotated in the 'streamline' coordinate system [34] before storage in a daily data file.

#### 3.2.2. Micrometeorological Station

It is equipped with standard meteorological sensors and soil data sensors (data are collected by a dedicated Campbel CR1000-X datalogger).

Standard sensors (2 m height):

**Cup anemometer** (Campbell A100R), **windvane** (Campbell W200P), **thermohygrometer** (Rotronic MP100), **four-components net radiometer** (Hukseflux NR01), **precipitation gauge** (EM ARG100), **pressure sensor** (Campbell PTB101B).

Soil sensors:

**2 thermistor temperature sensors** (Campbell109L, 2 and 5 cm depth), **2 thermopile soil heat flux sensors** (Hukseflux HFP01, 2 and 5 cm depth), **2 moisture content capacitive sensors** (Decagon EC-5, 2 and 35 cm depth).

A selection of the available IMS data can be visualized at the address [18] in graphic format. Data are freely available upon request at the address indicated in the staff area. Part of the dataset is also available in the Hymex site for subscribed users [35].

*3.3. Surface Water Budget*

For the surface water budget, in order to estimate the net infiltration, the following mass conservation equation is considered [22,36,37]:

$$S(t + dt) = S(t) + P - E - I - R + Ir, \tag{1}$$

where $S(t)$ is the soil water content per unit area, and $P$, $E$, $I$, $R$, and $Ir$ are the total precipitation, actual evapotranspiration, net infiltration, surface-subsurface runoff, and irrigation water in the time interval $dt$, respectively. Due to the hydrological features of the measurement site and its surroundings, the runoff term $R$ has been neglected. In fact, the site is placed on a large sub-horizontal ground without any appreciable slope, thus neither stream systems nor persistent pond waters are present.

To quantify the surface water transfer data from the IMS the following data have been used:

(1) total precipitation $P$;
(2) averaged evapotranspiration flux;
(3) averaged soil moisture at 2 cm ($s1$) and 35 cm ($s2$) depth.

All data are taken on a 30 min base interval (averages, and totals for precipitation). The averaged evapotranspiration flux has then been transformed to 30-minutes total evapotranspiration $E$.

After calculating the total daily/monthly values for the above quantities, the daily/monthly net infiltration $I$ has been calculated by the following expression between the initial $(t)$ and final $(t + dt)$ time of each day/month [22]:

$$I = P - E - S(t + dt) + S(t), \tag{2}$$

In Equation (2) the $Ir$ term has been neglected (as well as $R$). The anthropic irrigation is indeed absent in the measurement area but still present over the whole MACES region and will be the object of further studies. In Equation (2) $P$ and $E$ are directly derived by the measurements, while $S(t)$ has been calculated approximating an average soil moisture content per square meter as:

$$S(t) = h(s1(t) + s2(t))/2, \tag{3}$$

where $s1$ and $s2$ are the available soil moisture density measurements (in $m^3/m^3$ at 2 cm and 35 cm underground, respectively) and $h$ represents an effective soil thickness estimated as approximately 0.4 m [22]. Negative values of $I$ (Equation (2)) are still possible in very dry conditions because of the approximations in Equation (3) but they must also be considered, as vanishing contributes to the MACES water balance (see also [22]). In the daily/monthly averaging of the evapotranspiration, where it is more likely to have some missing data, the actual percentage of the available data has also been calculated, generally resulting in over 80%. The results have then been corrected, adding the product of the lacking data percentage times of the calculated daily/monthly average to each total value, to approximately compensate for the effect of the missing data. The estimated infiltration for MACES is shown in Figure 3 with the related linear trend. A more complete data view can be found in "S2.txt" and "S3.txt" in Zenodo repository (https://zenodo.org/record/7572140).

## 4. User Notes

The trend of the water-table fluctuations (Figure 2) apparently suggests a decline of the groundwater resource and is in agreement with the negative trend in the estimated infiltration for the considered five hydrological years (Figure 3). The infiltration decrease is associated with a decrease in precipitation in the considered five years, while the real evapotranspiration shows a less significant negative trend in the same period (see also Appendix C). Thus, the reduced surface water input affects the MACES groundwater input. This finding gives an insight into the natural stresses that affected MACES from 2017 to 2022. However, no data are available regarding the shallow groundwater exploitation for

domestic and irrigation uses [21]. Moreover, the recharge of MACES by irrigation water (*Ir*) withdrawn from the CA during the warm/dry season is unknown. This currently prevents the calculation of the actual aquifer water budget. The measurements in the site herein shown and the sharing of the updated datasets are scheduled to continue for several more years.

**Author Contributions:** Conceptualization, methodology, investigation, data curation, writing, and editing, M.D.R. and P.M. All authors have read and agreed to the published version of the manuscript.

**Funding:** This research received no external funding.

**Institutional Review Board Statement:** Not applicable.

**Informed Consent Statement:** Not applicable.

**Data Availability Statement:** All data sets presented in this study are available in the Zenodo repository as text files. Additional data from the IMS data base (raw, 30-minutes averages) can be also freely requested to p.martano@isac.cnr.it.

**Acknowledgments:** The authors thank Luca Ciricugno (ISAC-CNR) for his technical support in the water-table measurement and Fabio Grasso (ISAC-CNR) and Cosimo Elefante (University of Salento) for their contributions to the management of the micrometeorological station data archive.

**Conflicts of Interest:** The author declare no conflicts of interest.

## Abbreviations

The following abbreviations are used in this manuscript:

| | |
|---|---|
| ISAC | Institute of Atmospheric Sciences and Climate |
| CNR | National Research Council |
| MACES | Miocene Aquifer of Central-Eastern Salento |
| IMS | ISAC-CNR Micrometeorological Station |
| COVID-19 | Coronavirus disease 2019 |
| CA | Cretaceous Aquifer |
| Hymex | Hydrological Mediterranean Experiment |
| UMTS | Universal Mobile Telecommunications System |
| FW | Fiorini Well |
| BW | Benessere Well |
| EI | Ente Irrigazione (agency for irrigation) |
| a.s.l. | above sea level |

## Appendix A. Hydrogeological Framework

The hydrogeological framework of the Salento Peninsula is characterized by two superimposed water bodies: the regional deep CA (permeable by fracturing and karst) and a series of local shallow aquifers (permeable by porosity, fracturing, and karst) [24–26]. The former is constituted of limestone and dolostone, the latter of porous calcarenite and partially lithified sand, all cracked by tectonic stress and affected by karst dissolution (with various degrees of impact on the hydraulic properties). The shallow MACES is located at the eastern Salento, where it constitutes the main water resource. The measurement site is close to the western boundary of MACES (Figure A1).

In the area of the measurement site, an aquiclude, constituted of Tertiary very low-permeability rocks [21], is interposed between MACES and CA. As a local result, the first is unconfined, while the second is here a confined water body (whose groundwater is under pressure greater than atmospheric). Regarding the hydraulic relationship among MACES and CA, the well EI 61/II (see Figure A2 for its location) drilled and equipped in the mid-1950s by the regional agency for irrigation (Ente Irrigazione, EI) is of reference. Under static conditions (no exploitation of groundwater), the aquifers showed the same water-table level (2.8 m a.s.l.) [21]. Such a result was confirmed by a survey performed in the mid-1980s [27], the last hydrogeological study carried out in this area.

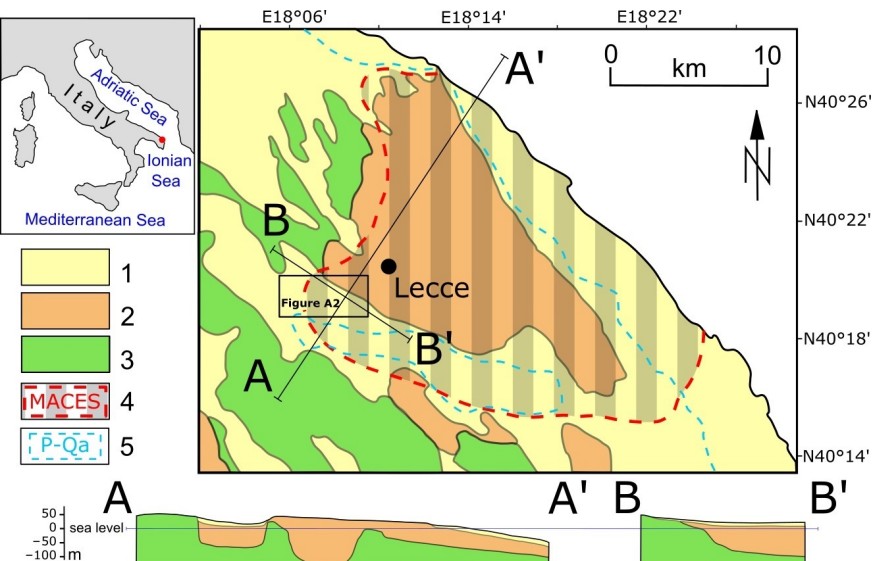

**Figure A1.** Geological map of the central-eastern Salento. 1. Plio-Quaternary units; 2. Miocene and Oligocene units; 3. Cretaceous units; 4. boundary of Miocene Aquifer of Central-Eastern Salento (MACES); 5. boundary of Plio-Quaternary aquifers; (after [21], modified).

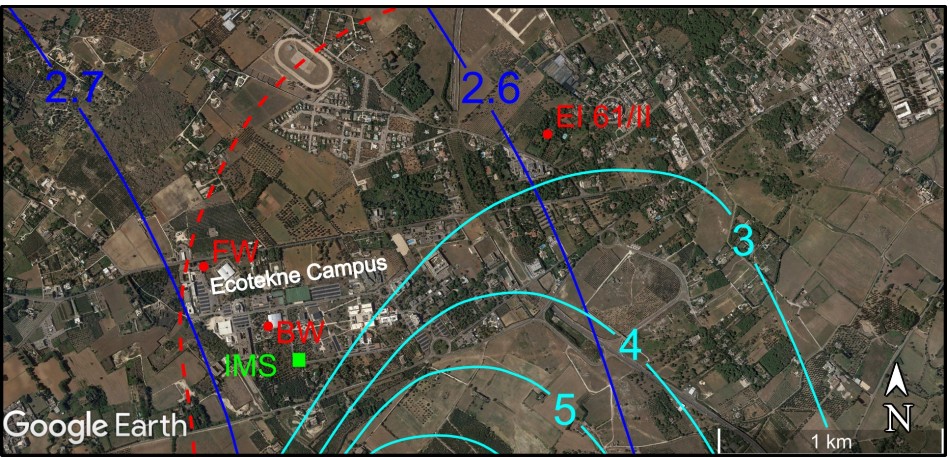

**Figure A2.** Contour map of the groundwater levels in the study area (m a.s.l.); solid light-blue line, Miocene Aquifer of Central-Eastern Salento (MACES); red dashed line, MACES boundary; solid blue line, regional Cretaceous Aquifer (CA); IMS, ISAC-CNR Micrometeorological Station; FW, Fiorini well; BW, Benessere well; EI 61/II, Ente Irrigazione well (after [21,27], modified).

Again with reference to the measurement site, MACES is supplied by water coming from a Plio-Quaternary aquifer that is superimposed on it at the southern boundary (Figure A1). The water exchanges between these aquifers are unknown but they should not significantly affect the water-table depths [27].

CA is in hydrodynamic equilibrium with intruding marine waters; thus, its water-table depth is determined by the Dupuit–Ghyben–Herzberg equation [38]. As such, it is affected by salinization processes [25,39]. The above equation may be written as follows:

$$Zg_s = (H + Z)g_f \tag{A1}$$

where $Z$ is the depth of freshwater below sea level, $H$ is elevation of water table above sea level, and $g_s$ and $g_f$ are the specific gravity of seawater and freshwater, respectively. If $g_s = 1.025$ and $g_f = 1$, then $Z = 41H$.

In the area of the measurement site, the water-table depth of CA is the groundwater base-level for MACES. Northwestward of the measurement site, the aquiclude gradually loses

the property to confine the aquifers for the reduced thickness and a variation in hydraulic properties of the Tertiary rocks; thus, the water of MACES drains into CA (cf. Figure A2).

**Appendix B. Data Sheet of the Monitoring Wells**

Well heads of FW and BW are cased by stone walls, circular in shape below the ground, and square in shape above the surface. The water-table measurements are taken by placing a 2 cm thick rod between two opposite sides of the stone casing (Figures A3 and A4).

The measurement scheme is shown in Figure A5. The height of the well heads was extracted by the provincial technical map at a scale of 1:5000 (Figure 1). More accurate altitude measures can be obtained by topographic leveling.

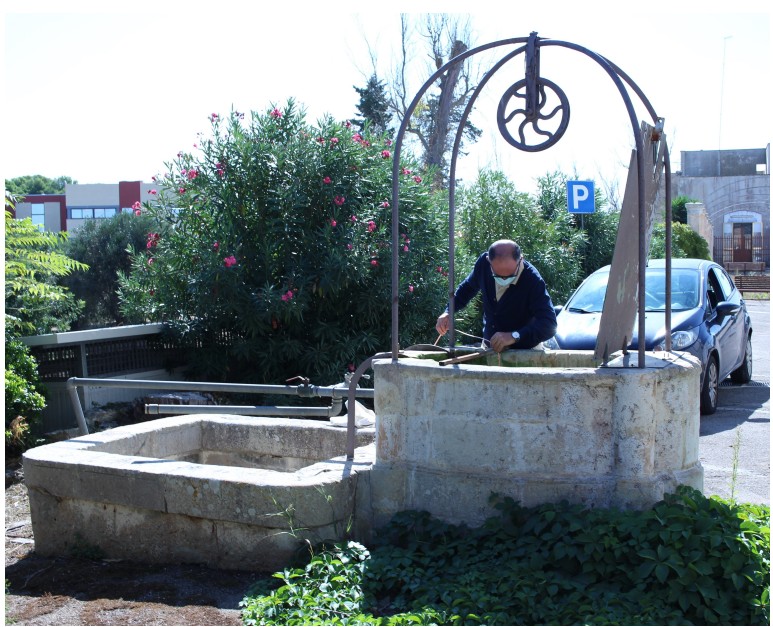

**Figure A3.** A measure of water-table depth in the Fiorini Well (note the rod placed between the opposite side of the stone wall to have a fixed reference point). Photo by Marco Delle Rose.

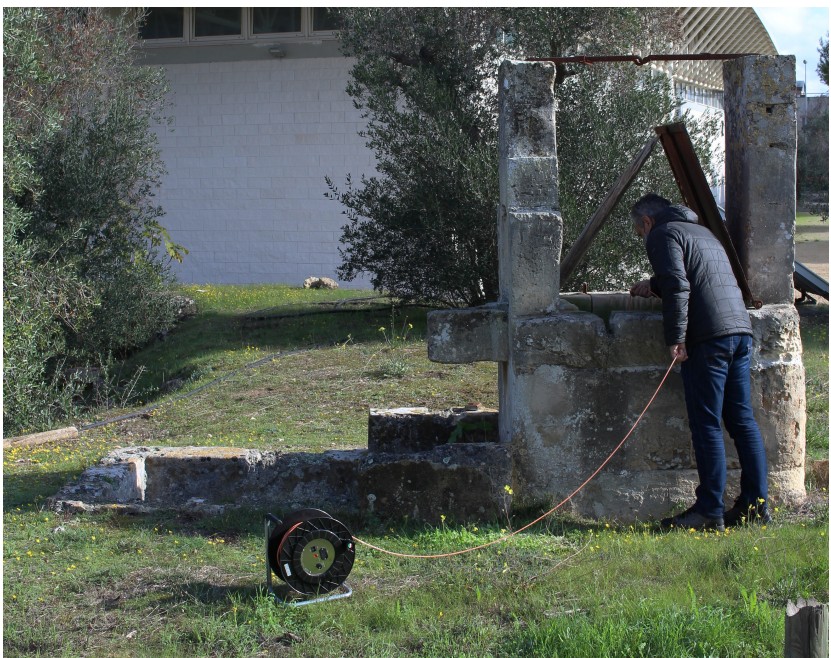

**Figure A4.** The descent of the electronic meter in the Benessere Well. Photo by Luca Ciricugno.

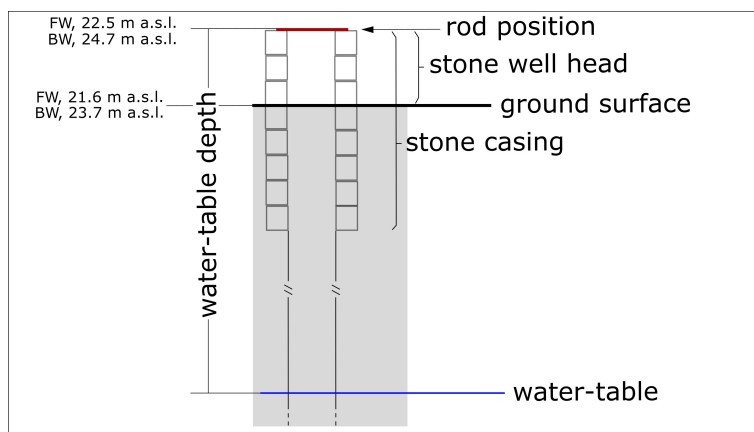

**Figure A5.** Scheme for the measurement of the water-table depth.

### Appendix C. Monthly Precipitation, Evapotranspiration, and Soil Moisture

Figure A6 shows a bar plot of the monthly total precipitation and evapotranspiration, as reported in the dataset. The clear decrease in precipitation in recent years does not correspond to a comparable decrease in the evapotranspiration.

Figure A7 shows the soil moisture at two levels (end of month data) as reported in the dataset. Note the almost total loss of water in the surface level in summer months compared with the retaining of water in the bulk soil data. The accumulated/lost water amount in the soil moisture profile between the beginning and the end of each period and its uncertainty affect the estimated infiltration.

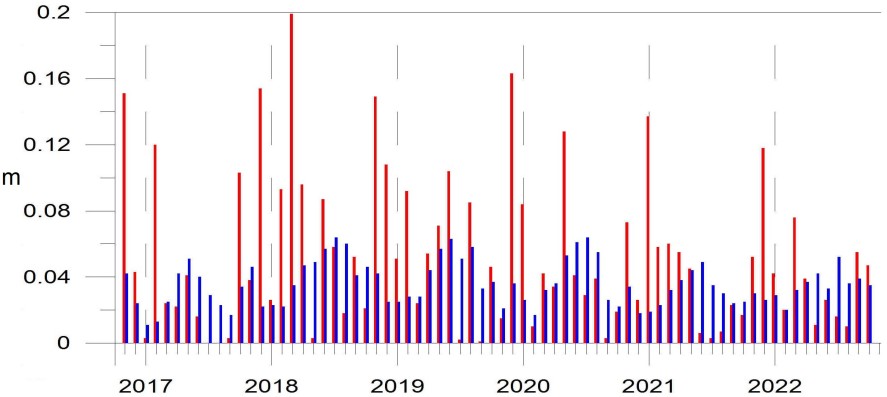

**Figure A6.** Bar plot of the total monthly precipitation (red) and evapotranspiration (blue) in meters.

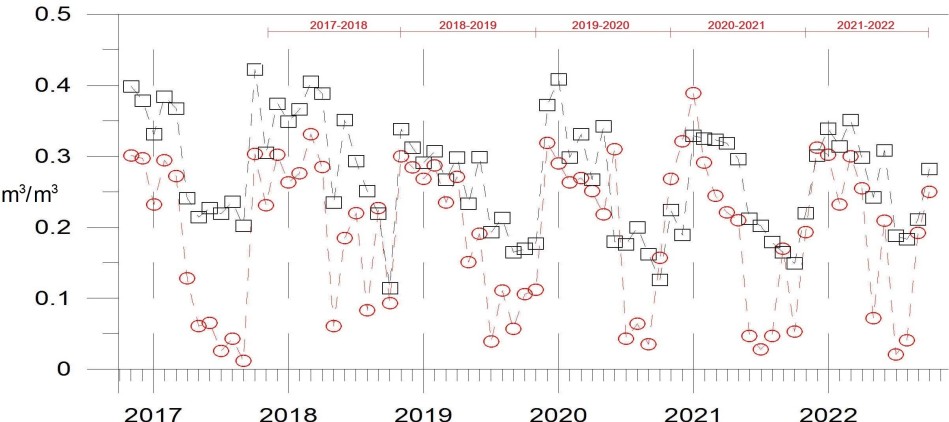

**Figure A7.** Plot of the soil moisture at 2 cm (red circles) and 35 cm depth (black squares), in $m^3/m^3$ (end-of-month data). The dashed lines are intended only to aid viewing. In the upper side, the considered hydrological years are delimited.

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
