# Peer review of "Datasets of Groundwater Level and Surface Water Budget in a Central Mediterranean Site (21 June 2017–1 October 2022)"

_data, 2017_

Round 1

Reviewer 1 Report

The paper by  Delle Rose and Martano “Multi-year datasets of groundwater level and surface water budget in a central Mediterranean site” introduces new data and describes its origin and processing. I recommend the paper for publication after minor corrections.

The Supplement contains three data files, but there is no README file, which is necessary to include (or equivalent documentation). It should contain brief description of datasets, authors, dates, and explanation of listed variables, which are listed in short form:

prec   

evtr  

soi1i   

soi1f   

soi2i   

soi2f   

infil 

percdata 

evtrc   

infilc

Guessing their meaning from the paper is not acceptable when publishing a dataset as the primary deliverable. Such data publication requires adequate documentation. 

I am not sure the title of the paper is suitable: I think “multi-year” should be replaced with specific time interval given in brackets at the end of the title.

It would be useful to include a Nomenclature for the multiple acronyms used in the paper.

Figure 2 has poor graphic resolution and absent variable and units in y-axis.

Figure 3 has no variable name and units in y-axis. Also, explain the dashed lines in the caption. This figure is equivalent to figure A7, and they should be of the same style.

The sentence in line 206 about anthropogenic stress is not clear, the authors have to clarify and expand this.

It is not clear why the important information on hydrogeology is placed in the appendix? I think it could be a section in the main text.

The Dupuit-Ghyben-Herzberg equation is mentioned in line 249, but I think it would be useful to include it as it is discussed in the following text.

Text in the appendix in lines 259-263 would be useful in the main text, where the wells are mentioned.

MINOR COMMENTS

In page1, line 27, “normal in…”

In page 2, line 53, “km” (not “kn”)

Check use of “Ecotekne” and “Ecotecne” across the manuscript

In page 3, lines 94-106: format the list of variables better, so it would be easy to see how many variables (use boldface, for example).

In page 4, line 113: “about every three days”

In page 4, line 114: remove double “see”

In page 4, line 116: “funding of research”

In page 5, line 164, there should be no indent in the first line after equation. The same in page 6, line 186.

In page 6, line 202: “associated with”

Abbreviation a.s.l. (above sea level) is not explained.

When speaking about monthly precipitation, there is no need to use term “cumulative”, because these are obviously aggregated values. Cumulation is not the same as aggregation, by the way.

Remove “final” from the caption of Figure A7.

“Accessed on” statement is incomplete in lines 312, 330, 335.

Author Response

dear Reviewer,

thank you for time and effort to review the manuscript.

General Comment: The paper by Delle Rose and Martano “Multi-year datasets of groundwater level and surface water budget in a central Mediterranean site” introduces new data and describes its origin and processing. I recommend the paper for publication after minor corrections.

Comment 1: The Supplement contains three data files, but there is no README file, which is necessary to include (or equivalent documentation). It should contain brief description of datasets, authors, dates, and explanation of listed variables, which are listed in short form:

prec

evtr

soi1i

soi1f

soi2i

soi2f

infil 

percdata

evtrc

infilc

Guessing their meaning from the paper is not acceptable when publishing a dataset as the primary deliverable. Such data publication requires adequate documentation.

Reply: As suggested by the editor, the data files have been stored in Zenodo.org.  A README file has now been added to the data files, containing a brief description of the contents and a detailed explanation of the listed variables .

Comment 2: I am not sure the title of the paper is suitable: I think “multi-year” should be replaced with specific time interval given in brackets at the end of the title.

Reply: The title has been modified according to the suggestion.

Comment 3: It would be useful to include a Nomenclature for the multiple acronyms used in the paper.

Reply: Abbreviations are now listed (see page 7).

Comment 4: Figure 2 has poor graphic resolution and absent variable and units in y-axis.

Reply: Previous Figure 2 has been replaced with a high graphic resolution new picture. Units have been placed in y-axis.

Comment 5: Figure 3 has no variable name and units in y-axis. Also, explain the dashed lines in the caption. This figure is equivalent to figure A7, and they should be of the same style.

Reply: The style of figure A7 has been changed to match that of figure 3.

The caption of Figure 3 has been improved with the required explanations, and the units have been added in the y axis of figs. 3 and 7.

Comment 6: The sentence in line 206 about anthropogenic stress is not clear, the authors have to clarify and expand this.

Reply: the sentence has been amended according to the suggestion.

Comment 7: It is not clear why the important information on hydrogeology is placed in the appendix? I think it could be a section in the main text.

Reply: Information on hydrogeology is placed in the appendix because the paper is a Data Descriptor, thus the only features directly connected with the data sets should be described in the main text. Such ancillary hydrogeological information simply supply basic notions for potential users.

Comment 8: The Dupuit-Ghyben-Herzberg equation is mentioned in line 249, but I think it would be useful to include it as it is discussed in the following text.

Reply: The equation is now reported in the text.

Comment 9: Text in the appendix in lines 259-263 would be useful in the main text, where the wells are mentioned.

Reply: The sentences has been moved in the main text (see page 4).

Minor Comments

In page 1, line 27, “normal in…”

In page 2, line 53, “km” (not “kn”)

Check use of “Ecotekne” and “Ecotecne” across the manuscript

In page 3, lines 94-106: format the list of variables better, so it would be easy to see how many variables (use boldface, for example).

In page 4, line 113: “about every three days”

In page 4, line 114: remove double “see”

In page 4, line 116: “funding of research”

In page 5, line 164, there should be no indent in the first line after equation. The same in page 6, line 186.

In page 6, line 202: “associated with”

Abbreviation a.s.l. (above sea level) is not explained.

When speaking about monthly precipitation, there is no need to use term “cumulative”, because these are obviously aggregated values. Cumulation is not the same as aggregation, by the way.

REPLY: The term ‘cumulative’ has been dropped or substituted with ‘total’.

Remove “final” from the caption of Figure A7.

REPLY: In the caption of figure 7  ‘final monthly’ has been properly substituted with ‘end of the month’.

“Accessed on” statement is incomplete in lines 312, 330, 335.

General reply to minor comments: all minor comments were considered and related changes made; a.s.l. is now listed in “Abbreviations” at page 7.

Reviewer 2 Report

Overall there are no significant points to make.

A couple of mistakes:

line 51 - 150 km;

line 121 - dd/mm/yy;

Figure A1 - Quaternary

In addition in refs 18, 27 and 30 is missing de date of access

Author Response

dear Reviewer,

thank you for time and effort to review the manuscript.

General Comment: Overall there are no significant points to make.

Reply: Thanks for your positive review.

Minor comments:

A couple of mistakes:

line 51 - 150 km;

line 121 - dd/mm/yy;

Figure A1 - Quaternary

In addition in refs 18, 27 and 30 is missing de date of access.

Reply to minor comments: all comments were considered and related changes made

Reviewer 3 Report

Review for data-2127205 ‘Multi-Year datasets of groundwater level and surface water budget in a Central Mediterranean site

The manuscript provides data description on the monitoring of groundwater levels and on water budget modeling in Mediterranean.

The data is original and very useful for water resource planning and management. Better description on the data and method will improve the readability and replicability of the manuscript.

·       L20 ‘parameters’ change to ‘variables’

·       L48 geophysical setting?

·       L51 I would not say ‘precipitations are usually limited…’ but better phrase such as ‘… precipitation … % lower than surrounding area in Mediterranean…’ or other phrases.

·       Please provide province/region and country for the location of measurement.

·       L73-74 how the depth of groundwater levels in the study site differ from literature?

·       Figure 1. please provide country, where the measurement is

·       Section 2.2.1 and Section 2.2.2. Meteorological station. I think this is not part of Data descriptor as the authors didn’t measure weather variables.   

·       Section 2.3

o   I suggest the authors to use strong words/phrases that the groundwater levels were manually measured using water table meter.

o   The frequency of measurement was haphazard as shown in Table S1. Better explain on this issue will improve the readability of the manuscript.

·       Section 2.4 Data records

o   Please clearly indicate the period date of measurement  (L120)

o   L123 gg to dd

o   Figure 2:

§  the x-axis label shall be in the bottom panel.

§  The unit for y-axis

o   Caption of Figure 3: delete groundwater. If use ‘groundwater infiltration’, does it differ from groundwater recharge?

·       L157 water-table to groundwater levels for consistency with Section 2.3

Author Response

dear Reviewer,

thank you for time and effort to review the manuscript.

General comment: The manuscript provides data description on the monitoring of groundwater levels and on water budget modeling in Mediterranean. The data is original and very useful for water resource planning and management. Better description on the data and method will improve the readability and replicability of the manuscript.

Reply: We have improve the readability and replicability of the manuscript according your comments. Also, a README file has been added to the datasets. The title has been slightly changed based on a comment of a reviewer. Further amendments have been made based on the suggestions of other reviewers.

Comment 1: L20 ‘parameters’ change to ‘variables’

Reply: The change has been made.

Comment 2: L48 geophysical setting?

Reply: The change has been made.

Comment 3: L51 I would not say ‘precipitations are usually limited…’ but better phrase such as ‘… precipitation … % lower than surrounding area in Mediterranean…’ or other phrases.

Reply: The sentence has been rewritten and improved.

Comment 4: Please provide province/region and country for the location of measurement.

Reply: Please, see L34 “(Apulia region, Italy)”.

Comment 5: L73-74 how the depth of groundwater levels in the study site differ from literature?

Reply: the depth of groundwater levels in the study site is in good agreement with literature data; see new L74.

Comment 6: Figure 1. please provide country, where the measurement is

Reply: Region and country are now in the caption.

Comment 7: Section 2.2.1 and Section 2.2.2. Meteorological station. I think this is not part of Data descriptor as the authors didn’t measure weather variables.

Reply: Sections 2 and 3 have been rearranged and improved

Comment 8: Section 2.3. I suggest the authors to use strong words/phrases that the groundwater levels were manually measured using water table meter. The frequency of measurement was haphazard as shown in Table S1. Better explain on this issue will improve the readability of the manuscript.

Reply: we have changed the title of the section and added "Manual" at the beginning of the second paragraph (L115) to meet the suggestion. New sentences (L118-121) has been inserted to better explain the choice of the frequency of measurement. This frequency agrees with those used in the literature.

Comment 9: Section 2.4 Data records. Please clearly indicate the period date of measurement  (L120). L123 gg to dd

Reply: the period date is now clearly indicated. “gg” has been replaced with “dd”.

Comment 10: Figure 2: the x-axis label shall be in the bottom panel. The unit for y-axis.

Reply: These corrections have been made. Moreover, we have improved the graphic resolution of Figure 2.

Comment 11: Caption of Figure 3: delete groundwater. If use ‘groundwater infiltration’, does it differ from groundwater recharge?

Reply: The term ‘net infiltration’, defined in eq. (2), is now used everywhere in the manuscript.

Comment 12: L157 water-table to groundwater levels for consistency with Section 2.3.

Reply: The sentence has been rewritten.

Round 2

Reviewer 3 Report

Dear authors,

Thanks for your efforts to improve the manuscript. Below are more some comments on the revised manuscript.

1. Please be consistent, in abstract its said micrometeorological station, but in the method it was meteorological station. 

2. I would not use term of 'a significant decreasing trend (L8)' if the statistical analysis is not available. other phrase such as 'substantial' may work. 

3. L142 groundwater level measurement

4. the keywords may too many

Best

Author Response

Thanks for the additional comments. We have made the suggested changes. "Mediterranean climate" has been deleted from the keywords. Please see attached file.
